# Customized Multichannel Measurement System for Microbial Fuel Cell Characterization

**DOI:** 10.3390/bioengineering10050624

**Published:** 2023-05-22

**Authors:** Nicola Lovecchio, Valentina Di Meo, Andrea Pietrelli

**Affiliations:** 1Department of Information Engineering, Electronics and Telecommunications, Sapienza University of Rome, Via Eudossiana 18, 00184 Rome, Italy; andrea.pietrelli@univ-lyon1.fr; 2Institute of Applied Sciences and Intelligent Systems, National Research Council of Italy, Via Pietro Castellino 111, 80131 Naples, Italy; valentina.dimeo@na.isasi.cnr.it; 3Univ Lyon, INSA Lyon, Universite Claude Bernard Lyon 1, Ecole Centrale de Lyon, CNRS, Ampere, UMR5505, 69621 Villeurbanne, France

**Keywords:** MFC measuring system, customized electronics, MFC characterization, automatic measurements, microbial fuel cell, sigma-delta analog-to-digital converter, transimpedance amplifier, dual-channel board

## Abstract

This work presents the development of an automatic and customized measuring system employing sigma-delta analog-to-digital converters and transimpedance amplifiers for precise measurements of voltage and current signals generated by microbial fuel cells (MFCs). The system can perform multi-step discharge protocols to accurately measure the power output of MFCs, and has been calibrated to ensure high precision and low noise measurements. One of the key features of the proposed measuring system is its ability to conduct long-term measurements with variable time steps. Moreover, it is portable and cost-effective, making it ideal for use in laboratories without sophisticated bench instrumentation. The system is expandable, ranging from 2 to 12 channels by adding dual-channel boards, which allows for testing of multiple MFCs simultaneously. The functionality of the system was tested using a six-channel setup, and the results demonstrated its ability to detect and distinguish current signals from different MFCs with varying output characteristics. The power measurements obtained using the system also allow for the determination of the output resistance of the MFCs being tested. Overall, the developed measuring system is a useful tool for characterizing the performance of MFCs, and can be helpful in the optimization and development of sustainable energy production technologies.

## 1. Introduction

The detrimental impact of greenhouse gases, toxic emissions, and air pollution resulting from fossil fuel usage poses a growing threat to the environment and human health, necessitating an immediate shift towards renewable and sustainable energy alternatives [1]. In this context, Microbial Fuel Cells (MFCs) represent a promising technology for sustainable energy generation [2,3,4,5,6]. They offer a clean and cost-effective energy source for practical applications, making them attractive to researchers and engineers worldwide [7,8,9,10]. MFCs can convert chemical energy from organic compounds into electrical energy through the catalytic action of microorganisms. This is achieved by exploiting the electron transfer mechanisms of microorganisms, which enable them to transfer electrons from organic matter to an electrode [11,12,13]. As a result, MFCs can generate electrical power continuously, without the need for external energy sources, such as sunlight or wind, which can be intermittent and unstable [14,15].

In recent years, the interest in MFCs has grown rapidly due to their potential to provide sustainable energy source. These cells have been tested in a range of practical applications, such as wastewater treatment [16,17,18], desalination [19], and energy production [20,21]. MFCs have the potential to provide a clean, reliable, and cost-effective energy source, making them an attractive option for a range of practical applications.

The ability to analyze MFCs is critical for the design of optimized power management systems that can exploit the energy delivered by these cells [22]. Researchers are interested in developing new materials and methods to improve the efficiency of MFCs and reduce their costs [23,24,25,26]. One promising approach is to arrange MFCs in stacks or clusters to form low-power micro-grids [27,28,29], providing sustainable energy source for a wide range of applications, including robotics [30], lighting [31], and wireless sensor networks (WSNs) [32,33,34,35].

In addition to their potential as clean energy source, MFCs can also function as bioremediators in wastewater treatment plants (WWTPs) [36,37,38]. They can break down pollutants in the wastewater, removing the need for costly and energy-intensive treatment methods [39,40,41,42]. Alternatively, MFCs can produce bio-hydrogen instead of electricity, which can be used as fuel or feedstock for other industrial processes [43,44,45,46]. Moreover, MFCs can be used directly as biosensors [47] to monitor specific parameters affecting system performances, such as organic matter concentration [48], temperature [49], pH [50], biological oxygen demand (BOD) [51], and water quality [52]. This enables researchers to track changes in these parameters over time and optimize MFC performance accordingly.

One key challenge in exploiting the full potential of MFCs is understanding their dynamics and electrical performances. To address this issue, this paper presents a novel measuring system for the electrical characterization of MFCs based on our previous work [53]. In particular, the system is designed to be portable, easy to use, and low-cost, making it accessible to researchers and engineers worldwide. It allows for automated and long-term testing, providing key information about several parameters, including the internal resistance value, maximum power transfer point, and voltage, current, and power values under different conditions.

To the best of our knowledge, there are currently no similar systems reported in the literature, making our proposed device a unique and innovative solution for the characterization of MFCs. In this context, the measuring system here presented represents a valuable tool for the analysis and optimization of MFCs, and has the potential to accelerate the development of this promising technology towards practical applications in sustainable energy generation and wastewater treatment.

The paper is organized as follows: Section 2 describes the design of the proposed measuring system, Section 3 shows the developed prototype and software, while in Section 4 the obtained experimental results are reported together with some measurement examples. Finally, conclusions are drawn in Section 5.

## 2. Measuring System Design

The presented system is capable of measuring from 2 to 12 MFCs in parallel, making it an ideal tool for studying their performances under various conditions.

To achieve this, we developed a printed circuit board (PCB) that comprises a microcontroller for hardware management (hereafter referred to as the “Microcontroller Board”) and a dual-channel board (hereafter referred to as the “Front-End Board”) that allows for the simultaneous measurement of two MFCs. The system is also expandable, as the Microcontroller Board can drive up to six dual-channel boards, allowing the measurement of up to 12 MFCs in parallel.

In order to properly design the electronic circuits included in the system, we have considered the usual ranges of electrical parameters that can be obtained by a MFC. In particular, typical MFCs provide an open circuit voltage in range of 0.3–0.9 V and a power range from 1 to 2000 mW/m^2^ [54]. Moreover, the possibility to characterize clusters of cells, connected in series and/or in parallel to increase output current and/or voltage [55], has been taken into account.

### 2.1. Microcontroller Board

The Microcontroller Board is the main component of the measuring system, responsible for managing the hardware and communicating with a PC through a USB connection. It features a Microchip PIC18F4550 microcontroller, which is connected to the Front-End Boards using header connectors that allow for easy assembly of all the PCBs composing the system.

In addition to its communication functions, the Microcontroller Board includes an automatic voltage recognition system that can detect the power supply voltage, which can come from either the USB or an external 5 V adapter. This system is implemented through a circuit that includes a Double-Pole, Double-Throw (DPDT) relay (V23079-D2001-B301 from TE Connectivity) and an n-channel MOSFET (NX3008NBK from Nexperia). One pole of the relay is used to switch between the two power sources, with its normally closed (NC) contact connecting the USB voltage to the system’s power supply $V_{sys}$ and its normally open (NO) contact connecting the external voltage to the system. The other pole, instead, is used to switch between a green LED and a red one to visually indicate which is the effective power supply source.

To control the relay, its coil is connected to the external voltage coming from the 5 V adapter (V_EXT_) and to the transistor M_1_ as shown in Figure 1a.

The J_EXT_ connector, used for connecting the system to the external power supply, has a sleeve/shunt function that allows for automatic switchover from the USB voltage (V_USB_) to V_EXT_ when the plug is inserted or withdrawn. When the external plug is not inserted, the pin 1 of the J_EXT_ is connected to pin 2 (ground), and M_1_ is turned off. When the external plug is inserted, pin 1 is automatically disconnected from pin 2, and M_1_ turns on, being also connected to pin 3 (external 5 V) through the delay network R_1_ − C_1_. This delay network ensures that M_1_ turns on after about one-second delay to prevent the microcontroller from turning off during the power supply switch. Finally, the resistor R_EXT_ ensures that M_1_ turns off if an external plug from a non-functional power supply is inserted into J_EXT_. The maximum current provided by the USB allows for the expansion of the system using up to three Front-End Boards. To further expand the system, the external power supply is required.

Beyond the functions described above, the Microcontroller Board also generates a reference voltage ($V_{neg}$) of about 100 mV that is sent to the Front-End Boards. This voltage is generated by using a 2.5 V voltage reference (REF192 from Analog Devices), a resistive divider (1 kΩ–27 kΩ), and a couple of operational amplifiers (TS912 from STMicroelectronics), connected as shown in Figure 1b.

### 2.2. Front-End Board

The Front-End Board is responsible for the measurement of MFCs’ electrical parameters. Each board can measure two MFCs in parallel, allowing for the measurement of up to 12 MFCs using six Front-End Boards.

The board comprises two identical analog front-end circuits based on operational amplifiers for current measurements and two 16-bit sigma-delta analog-to-digital converter (ADC) for high-precision measurement of the MFCs’ voltage and current. The chosen ADC is the AD7706 (from Analog Devices), which is a three-pseudo-differential input channel converter. These ADCs (one per channel) communicate with the microcontroller placed on the Microcontroller Board through a serial peripheral interface (SPI).

The board can operate in three modes: open-circuit voltage measurement, short-circuit current measurement, and power output measurement obtained by discharging the MFC on a digital selectable resistor. The switching between the different modes is performed by two relays, one DPDT and one Single-Pole Double-Throw (SPDT), as shown in Figure 2.

In particular, Figure 2a illustrates the position of the relays and the circuit configuration in case of voltage measurement, Figure 2b shows the circuit configuration for short-circuit current measurement, and Figure 2c shows the circuit configuration for power output measurement. To control the integrated relays for changing the measurement mode, we have used the same transistor of the Microcontroller Board (NX3008NBK). Moreover, in order to control all the required MOSFETs integrated on the Front-End Board, we have chosen to use an eight-channel I/O expander with SPI interface (MCP23S08 from Microchip). In this way, control signals coming from the Microcontroller Board are strongly reduced, allowing the expansion of the system for measuring up to 12 MFCs in parallel.

#### 2.2.1. Open-Circuit Voltage Measurements

In open-circuit voltage measurement mode (Figure 2a), the positive terminal of the MFC is connected directly to the ADC, while the negative terminal is connected both to the reference voltage coming from the Microcontroller Board ($V_{neg}$) and to the converter. As mentioned above, the used ADC is a 16-bit sigma-delta converter with an internal programmable gain. The resolution of the ADC ($V_{res}$) can be calculated as:(1)$$V_{res}=\frac{V_{max}}{2^{16}-1}=\frac{V_{ref}}{\left(2^{16}-1\right)\cdot G_{ADC}}$$
where $V_{ref}$ is a reference voltage of 2.5 V coming from voltage reference REF192, $G_{ADC}$ is the internal gain of the ADC, and $V_{max}$ is the maximum voltage that can be measured by the converter and depends on the gain value.

For a gain of 1, the resolution is approximately 38.15 µV, and the maximum measurable voltage in this case is 2.5 V. As the gain is increased, the resolution becomes finer, but the maximum measurable voltage decreases proportionally. For example, at a gain of 2, the resolution is approximately 19.07 µV, and the maximum measurable voltage is 1.25 V. At a gain of 4, the resolution is approximately 9.54 µV, and the maximum measurable voltage is 625 mV. Finally, at a gain of 8, the resolution is approximately 4.77 µV, and the maximum measurable voltage is 312.5 mV. Considering that the negative terminal of the MFC is connected to a voltage of about 100 mV, the differential voltage ranges are approximately 2.4, 1.1, 0.5, and 0.2 V, respectively. It is worth noting that this ADC, for gains up to 8, provides a typical peak-to-peak resolution of 16 bits at a sampling rate of 50 Hz, as reported in its datasheet [56]. Furthermore, use of gains higher than 8 is not necessary for this particular application since the maximum measurable voltage would become too small.

#### 2.2.2. Short-Circuit Current Measurements

In short-circuit current measurement mode (Figure 2b), the MFC current is converted to a voltage $V_{x}$ using a transimpedance amplifier (TZA) implemented with the LMP7721 (from Texas Instruments), which is a sub-picoampere input bias current precision operational amplifier. This transimpedance amplifier converts the current to a voltage proportional to the current flowing through the MFC, according to the equation:(2)$$V_{x}=V_{ref}-R_{TZA}\cdot I_{MFC}$$
where $R_{TZA}$ is a 50 Ω 0.1% resistor. The voltage $V_{x}$ is then transformed using a subtractor amplifier, also based on operational amplifiers (TS912), which implements the following operation:(3)$$V_{out}=V_{os}-V_{x}=V_{os}-V_{ref}+R_{TZA}\cdot I_{MFC}$$
where $V_{os}$ is approximately 2.85 V. This subtracting operation has been implemented in order to better match the conversion range of the ADC, with a consequent increase of the system resolution. The complete analog front-end circuit that implements these operations (i.e., the “Current/Voltage Conversion” block in Figure 2) is reported in Figure 3.

Considering the circuit reported in the figure, the voltage $V_{os}$ can be expressed as:(4)$$V_{os}=2\cdot \frac{R_{x}\;+\;R_{os}}{2R_{x}\;+\;R_{os}}\cdot V_{ref}\;\approx \;2.85\;V$$

The signal $V_{out}$ is then sent to the ADC for the conversion, obtaining a resolution in terms of current measurement that can be calculated as:(5)$$I_{res}=\frac{V_{ref}}{\left(2^{16}-1\right)\cdot G_{ADC}\cdot R_{TZA}}$$

Considering gains of 1, 2, and 4, the current resolution is 0.763, 0.381, and 0.191 µA, respectively. For the calculation of the maximum current values that the system can detect, we have to consider that, for a MFC current equal to zero, the output voltage of the analog front-end circuit is $V_{os}-V_{ref}$, i.e., 350 mV. In this case, $I_{max}$ can be calculated as:(6)$$I_{max}=\frac{V_{max}-V_{os}+V_{ref}}{R_{TZA}}=\frac{V_{max}-350\;mV}{R_{TZA}}$$

Considering once again gain values of 1, 2, and 4, the maximum current values are equal to 43, 18, and 5.5 mA, respectively. The gain 8 has not been considered because it would lead to a negative value of $I_{max}$.

#### 2.2.3. Power Measurements

In power output measurement mode (Figure 2c), the board discharges the MFC on a selectable resistor controlled by the dual-channel Microchip MCP4241-103 digital potentiometer, as shown in Figure 2c. The current is then measured using the ADC and the TZA, and the MFC voltage $V_{MFC}$ and power output $P_{MFC}^{\;out}$ are calculated using the following formulae:(7)$$\left\{\begin{array}{c} V_{MFC}=R_{sel}\cdot I_{MFC}\\ P_{MFC}^{\;out}=V_{MFC}\cdot I_{MFC}=R_{sel}\cdot I_{MFC}^{2} \end{array}\right.$$
where $R_{sel}$ represents the selected discharging resistance. The MCP4241-103 allows a resistance $R_{sel}$ between 75 Ω and 10 kΩ with a step of 78.125 Ω to be selected, where 75 Ω represents the wiper resistance of the digital potentiometer.

### 2.3. Firmware and Software

To enable the communication between the measuring system and a PC, we developed a firmware for the Microcontroller Board and a software for the PC. The firmware allows for the configuration of the system’s parameters, such as the sampling rate and the number of MFCs to be measured, as well as the ranges of the voltage and current measurements.

The software, developed in Visual C++, enables the real-time visualization of measured data, including the MFCs’ voltage and current, as well as the voltage and the power of the cell as a function of the MFC current for different resistive loads. The software also allows for the export of data in a tab-delimited text format for further analysis.

In addition, the software provides the flexibility to customize the acquisition time-step, which is the time interval between two consecutive acquired data points. This feature allows users to optimize the data acquisition process depending on the type of measurement (voltage or current) being performed. Furthermore, we have included a variable time-step feature that allows to reduce the amount of data saved during long-duration measurements, which may last for hours or even days. This helps to avoid storing excessive amounts of data and simplifies the data analysis process. It is worth noting that the variable time-step feature not only helps to reduce the amount of data saved but also allows for the acquisition of the first charging and discharging phases of the MFCs with a high sampling rate. This is because, during the initial stages, the changes in voltage and current can occur very rapidly, and a non-constant time-step can capture these variations with greater precision, providing valuable insights into the MFCs’ behavior. Therefore, the software’s flexibility in customizing the acquisition time-step and implementing a variable time-step feature is crucial in ensuring the accurate measurement and analysis of the MFCs’ performance.

Finally, the software also allows for the creation of multi-step discharge protocols on different resistive loads, with the option of including recharging phases of the cell (voltage measurement in open-circuit condition) between discharging steps. This feature enables researchers and engineers to design and execute complex experiments with a variety of testing conditions and parameters to automatically infer, for example, the internal resistance of the MFC. The ability to customize and execute such protocols provides a comprehensive analysis of the MFCs’ behavior and performance under different scenarios. With this advanced functionality, the software offers an essential tool for conducting research and development in the field of microbial fuel cell technology.

## 3. Measuring System Development

Several software tools have been utilized to develop the whole measuring system. Altium Designer© was used to design the PCBs constituting the system’s physical hardware. MPLAB© IDE was used to write and compile the firmware code that runs on the microcontroller at the heart of the system. Finally, Microsoft© Visual Studio was used to develop the graphical user interface (GUI) that allows users to interact with the system.

### 3.1. Hardware

The realized PCBs were manufactured by using a standard two-layer FR-4 material with a thickness of 1.6 mm. Figure 4a shows the 3D model of the fabricated Microcontroller Board.

In particular, as highlighted in Figure 4a, this PCB is equipped with connectors H1 (18 pins), H2 (6 pins), and H3 (6 pins). These connectors provide all the required signals for controlling the Front-End Boards and comprise:the system power supply, including the 5 V and the ground lines;the SPI interface, including the data-in (SDI), data-out (SDO), and clock (SCK) signals;the negative voltage reference ($V_{neg}$);twelve chip-select (CS) signals (two for each Front-End Board) for the SPI communication with the integrated ADCs;six CS lines (one for each Front-End Board) to enable the SPI communication with the integrated digital potentiometers;six CS lines (one for each Front-End Board) for the SPI communication with the I/O expanders that allow to switch between the three measurement modes.

The 3D model of the fabricated Front-End Boards is reported in Figure 4b.

As previously explained, the measuring system is designed to drive up to six Front-End Boards that are connected to the Microcontroller Board. The Front-End Boards are differentiated by the channels they include, with the first board comprising channels 1 and 2, the second board comprising channels 3 and 4, and so on, up to the sixth board which includes channels 11 and 12. To properly connect the CS signals of the two ADCs, the digital potentiometer and the I/O expander of each board, four 0 Ω 0805-case resistors have been soldered onto the slots near the connectors H1, H2, and H3, corresponding to the channels managed by the board, as shown in the bottom view of the Front-End Board comprising channel 1 and channel 2, in Figure 5.

The measuring system components (Figure 6a) are enclosed in a black metallic box (Hammond Manufacturing 1550WFBK), as depicted in Figure 6b.

The box accommodates all the required boards to build a system capable of measuring up to six MFCs in parallel. Banana plug/alligator clip cables (Figure 6d) can be used to connect the MFCs-under-test to the system, as the box features banana sockets (Figure 6c).

### 3.2. Software

The measuring system’s software was designed to provide a user-friendly interface for configuring and acquiring data from the Microcontroller Board. The GUI is organized into several group boxes, each responsible for different functionalities, as shown in Figure 7.

The voltage and current measurements are handled by two identical group boxes, highlighted in blue and red in Figure 7, respectively. These boxes allow users to select the initial time step and set the measurement duration, either for a specified time or until manually stopped. Additionally, the software provides the option of a non-constant time step, which can increase linearly, quadratically, or exponentially with time, allowing users to optimize data acquisition depending on the measurement being performed.

For power measurements, the group box highlighted in yellow in Figure 7 allows users to select the resistance value. The group boxes highlighted in dark green and light green enable users to manually select the measurement mode and to program multi-step discharge protocols on several resistive loads, respectively. The latter feature also allows users to include recharging phases of the cell (voltage measurement in open-circuit condition) between discharging steps, enabling the analysis of the MFCs’ behavior under various scenarios.

Furthermore, the software allows to perform real-time data visualization of the measured voltage and current values as a function of time (“Real Time Measurements” tab in Figure 7), as well as the voltage and power of the cell as a function of the MFC current for different resistive loads (“Steady State Results” tab). The data can be saved and exported in tab-delimited text format for further analysis by simply selecting the check-box “Save Data”.

In addition to the features previously described, the software also provides the ability to calibrate the transimpedance amplifier on a channel-by-channel basis. This calibration process is essential to ensure accurate current measurements, as it compensates for any offset or gain errors introduced by the amplifier’s components.

To calibrate a channel, users need to provide a variable current generator. The software then measures and shows in real time the output voltage of the TZA (by selecting the check-box “Calibration Mode”) in order to calculate the corresponding transimpedance gain. The calculated gain is then loaded into the calibration window (Figure 8) to adjust the amplifier’s settings and to achieve accurate current measurements.

The calibration window also offers the ability to select the voltage and current ranges for measurements and the channels that are currently in use.

## 4. Experimental Results

In this section, we present the experimental results obtained from the measuring system described in the previous sections. First, we describe the calibration process, which is crucial for ensuring accurate measurements. We then present the results obtained from the voltage and current measurements, the power measurements, and the multi-step discharge protocols. In order to perform these tests, a set of battery packs were assembled to simulate the behavior of a real MFC.

### 4.1. Calibration of the Measuring System

To calibrate the transimpedance amplifiers, we used a variable current generator (Keithley 236 Source-Measure Unit) to provide a known current input to each channel. During the calibration process, the software displayed a real-time graph of the TZA output voltage in terms of the ADC readings, allowing us to associate each input current value with its corresponding ADC reading. Then, we performed a linear fitting on the obtained results to calculate the intercept and slope of the fitting line. Finally, we entered these values into the software’s calibration window to adjust the amplifier’s settings and achieve accurate current measurements. The nominal values of the intercept and slope (at the ADC gain equal to 1) can be calculated using the following equations:(8)$$I_{intercept}=\frac{V_{ref}-V_{os}}{R_{TZA}}\approx \;-7\;mA$$
(9)$$I_{slope}=\frac{V_{ref}}{\left(2^{16}-1\right)\cdot R_{TZA}}\approx \;7.6\cdot 10^{-4}\;mA$$

These equations can be derived from (Equation 3), considering that the voltage $V_{out}$ to be read from the ADC can be expressed as:(10)$$V_{out}=\frac{V_{ref}}{\left(2^{16}-1\right)}\cdot ADC_{reading}$$

We repeated this process for each channel of the system, and the obtained results are reported in Figure 9.

### 4.2. Voltage and Current Measurements

As a first step to evaluate the performances of the calibrated measuring system, we measured the current for two minutes without connecting any cells to the system. The purpose of this measurement was to assess the low noise level achievable with the sigma-delta architecture and the precision of the calibrated electronics at zero current.

Figure 10 shows the measured current for each of the six channels of the system as a function of time.

Each plot represents a 2-min acquisition with a sampling frequency of 10 Hz and an ADC gain of 4. Notably, all six channels exhibit a noise level within 2 bits of the least significant bit (LSB) of the 16-bit ADC, which is consistent with the noise level of the sigma-delta converter.

The measured current values are all very close to zero, within a mean value lower than ±0.1 µA for all channels. These results demonstrate the achieved low noise and high precision of the measuring system, as well as the effectiveness of the calibration procedure.

To further test the functionality of the measuring system, six battery packs were assembled using a 1.5 V AA battery, three resistors, and an electrolytic capacitor to simulate the behavior of a real MFC. The circuit diagram for each battery pack is shown in Figure 11.

As shown in Figure 11, each pack had the same component values except for the output resistance $R_{O}$, which was varied between 1 kΩ and 4.7 kΩ. To evaluate the performances of the measuring system under realistic conditions, the six battery packs were connected to the system, and their open-circuit voltage and short-circuit current were recorded over time. The results are reported in Figure 12, where Figure 12a shows the current measurement and Figure 12b shows the voltage measurement for the six channels.

The measurements were conducted sequentially, starting with a 10-s open-circuit voltage measurement as initial step, followed by a 30-s discharge measurement. As shown in Figure 12a, the current curves for each channel have different shapes due to the different output resistances used in the battery packs. This result confirms that the measuring system is capable of accurately detecting and distinguishing the current signals from different cells with varying output characteristics.

After the short-circuit current measurement, a 4-min open-circuit voltage measurement was performed for each channel. As expected, the voltage curves for all channels have the same behavior, since the internal capacitor charge in the battery packs is independent of the output resistance. The deviations observed in the measured curves are due to the tolerances of the components included in the battery packs.

### 4.3. Power Measurements

To further evaluate the performances of the measuring system, we have conducted power measurements using the same six battery packs as before except for the $R_{1}$ and $R_{2}$ values (Figure 11), that have been chosen equal to 1 kΩ and 470 Ω, respectively. The multi-step discharge protocol was applied to each pack with several resistive loads by changing the digital potentiomenter value ($R_{sel}$ of Equation (Equation 7)) every 10 s to obtain current and voltage measurements. The current measurements for all six channels are shown in Figure 13a.

The power output of each channel was automatically calculated by the software using the current and voltage measurements following the Equation (Equation 7), and the resulting current-power graphs are shown in Figure 13b. The six channels exhibit similar trends, but with different peak power outputs since these values are determined by the output resistances of the battery packs. The load resistance at which the power peak is located corresponds to the maximum power point transfer of the cell, where the load resistance is equal to the output resistance. Basically, these measurements allow us to determine the output resistance of the MFC being tested.

In particular, to precisely infer the cells’ output resistances, the power-current curves can be fitted by using a second order equation as:(11)$$P_{out}=-A_{1}\cdot I_{out}^{2}+A_{2}\cdot I_{out}$$

In this equation, the $A_{1}$ parameter value represents the resistance where the output power is maximum, i.e., the output resistance of the cell under test.

The results obtained from the fitting performed on the data in Figure 13b have been reported in Table 1 for all channels, together with the nominal values of the battery packs’ output resistances and the relative measurement errors expressed in percentage.

The nominal value of the output resistance of each battery pack is calculated using the formula:(12)$$R_{out,i}=R_{O,i}+\frac{R_{1}\cdot R_{2}}{R_{1}+R_{2}}$$
where $R_{1}$ and $R_{2}$ are equal to 1 kΩ and 470 Ω, respectively, while the $R_{O,i}$ resistances values are those reported in Figure 11.

As shown in Table 1, the errors that we found are all lower than ±10%. These measurements and elaborations demonstrate the ability of the measuring system to accurately measure power output from MFCs. The consistent results across all six channels further attest to the precision and reliability of the measuring system.

## 5. Conclusions

We have presented a multi-channel measuring system for monitoring the performances of microbial fuel cells (MFCs) in real-time. The system is based on low-noise sigma-delta analog-to-digital converters, which allow for high-resolution voltage and current measurements, and includes transimpedance amplifiers for accurate current sensing. We have described the calibration procedure used to adjust the amplifier settings and achieve precise measurements, and demonstrated the system’s ability to measure low-current signals with a high degree of precision.

To evaluate the measuring system performances, we conducted experiments on six battery packs with varying output resistances, simulating the behavior of real MFCs. The system was able to accurately detect and distinguish the current signals from different cells, and to measure the open-circuit voltage and short-circuit current over time. We also conducted power measurements using again battery packs, applying a multi-step discharge protocol to determine the output resistance and maximum power transfer of each cell. The results demonstrate the system’s ability to reliably measure power output from MFCs and to provide valuable insights into their performances.

Overall, the proposed measuring system represents a useful tool for MFCs research and development, allowing for real-time monitoring of their key performance parameters, such as voltage, current, and power. The system’s high resolution and accuracy, combined with its multi-channel analysis capability, make it suitable for a wide range of applications in the field of bioelectrochemical systems. Future works may focus on expanding the system’s capabilities, for example by incorporating wireless communication and remote data logging, or by developing algorithms for automatic detection of performance anomalies or optimization of operation parameters.

## Figures and Tables

**Figure 1 bioengineering-10-00624-f001:**
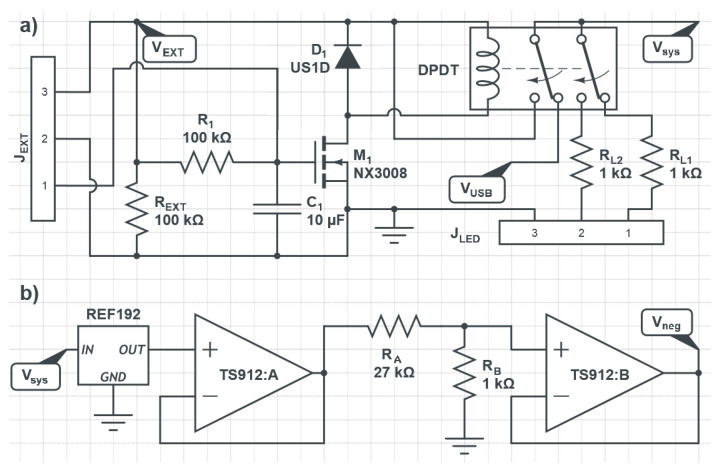
(**a**) Schematic of the automatic power supply voltage recognition system. (**b**) Schematic of the reference voltage $V_{neg}$ generation circuit.

**Figure 2 bioengineering-10-00624-f002:**
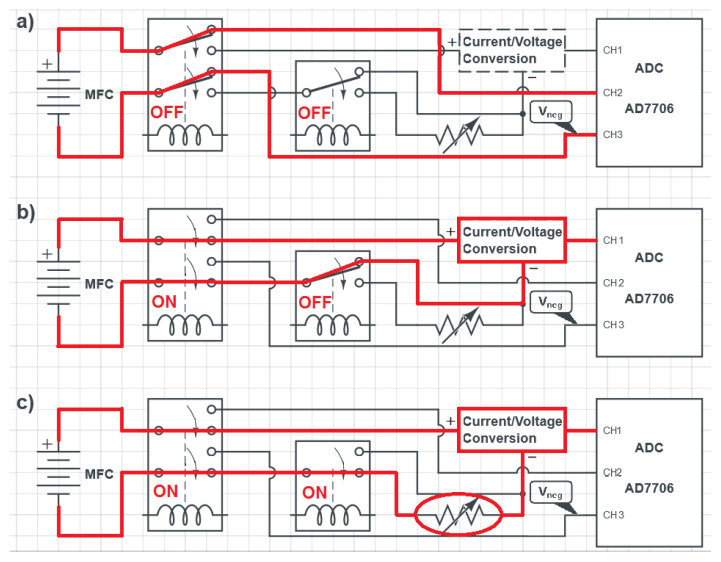
Block diagrams of the circuit for switching between measurement modes. The red line shows the circuit configuration for: (**a**) open-circuit voltage measurements, (**b**) short-circuit current measurements, and (**c**) power measurements.

**Figure 3 bioengineering-10-00624-f003:**
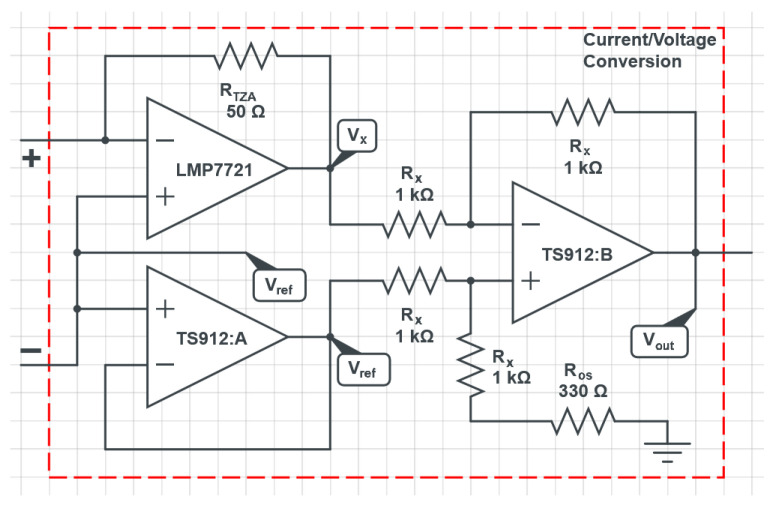
Schematic of the front-end electronics for the current/voltage conversion.

**Figure 4 bioengineering-10-00624-f004:**
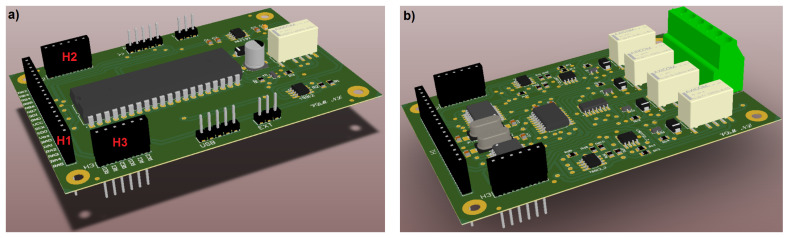
Three-dimensional models of the Microcontroller Board (**a**) and the Front-End Board (**b**).

**Figure 5 bioengineering-10-00624-f005:**
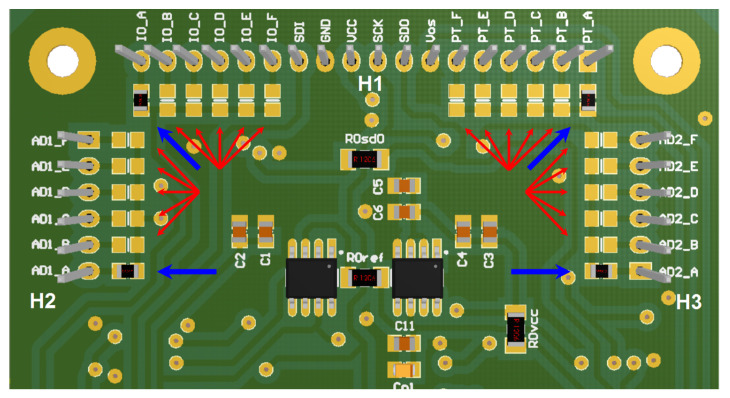
Bottom view of the 3D model of the Front-End Board. The picture shows the 24 slots where four 0 Ω resistors have been soldered to properly connect the CS signals for channels 1 and 2. Blue arrows indicate the soldered resistances, while red arrows show the free slots.

**Figure 6 bioengineering-10-00624-f006:**
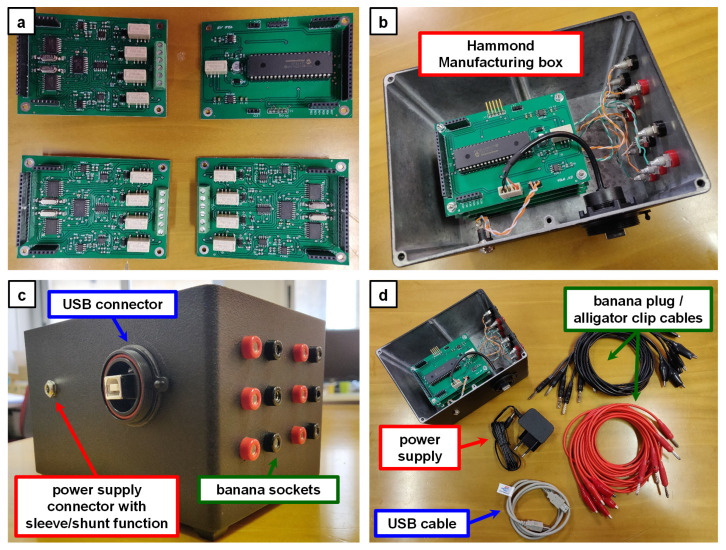
Pictures of the fabricated system. (**a**) One Microcontroller Board and three Front-End Boards before connections. (**b**) Stacked PCBs mounted in the metal box. (**c**) Box detail showing external panel connectors. (**d**) Overview of the system including cables and power supply.

**Figure 7 bioengineering-10-00624-f007:**
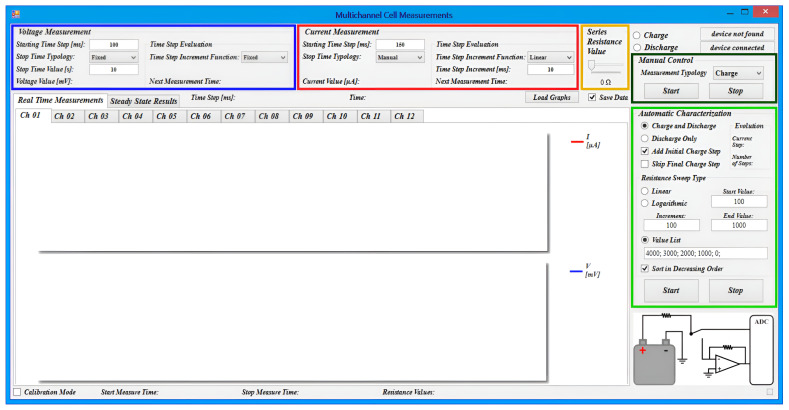
Graphical User Interface for controlling the acquisition system. Colored boxes in the figure highlight the main setting panels.

**Figure 8 bioengineering-10-00624-f008:**
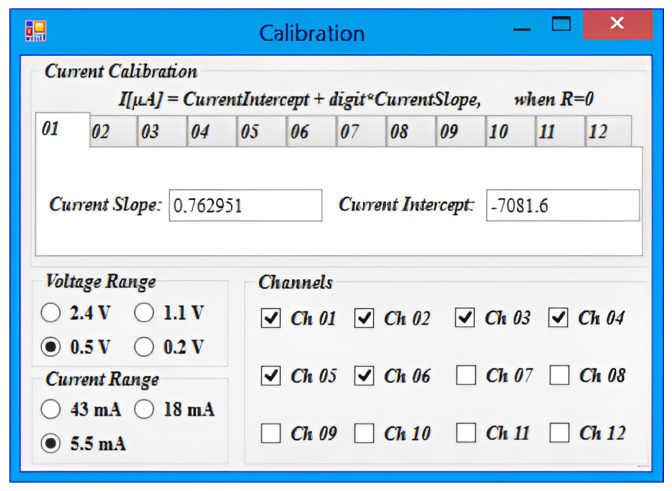
“Calibration” window, where the user can upload the current calibration values and can choose the voltage measurement range, the current measurement range, and the channels that are currently in use.

**Figure 9 bioengineering-10-00624-f009:**
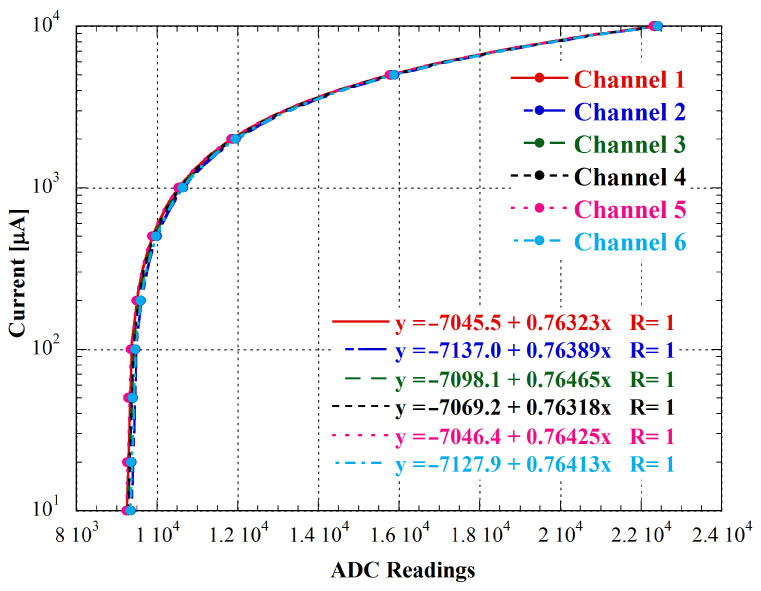
Calibration results. The current axis has been shown semi-logarithmically to better appreciate the linearity of the systems both for low and high values of current. Symbols represent the measured data, while colored equations and lines represent the linear fittings.

**Figure 10 bioengineering-10-00624-f010:**
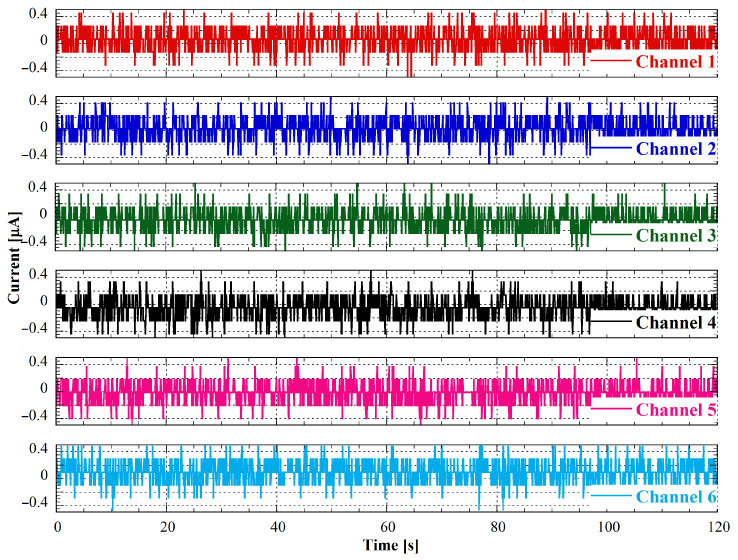
Zero-current measurements performed after the system calibration.

**Figure 11 bioengineering-10-00624-f011:**
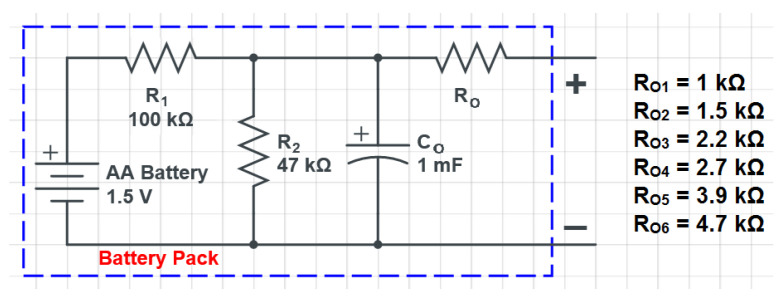
Battery pack circuit diagram. The resistances shown on the right of the circuit are those that differentiate the various battery packs.

**Figure 12 bioengineering-10-00624-f012:**
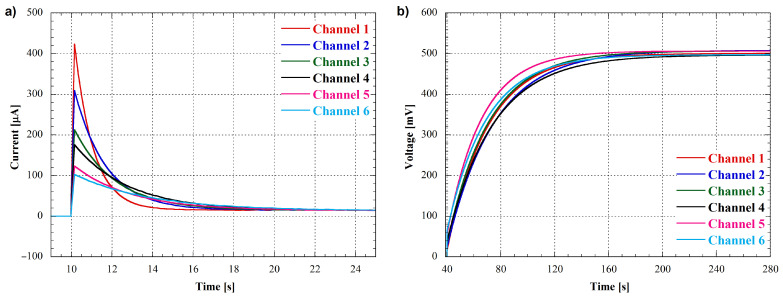
(**a**) Short-circuit current measurement of six battery packs. (**b**) Open-circuit voltage measurement of the six battery packs after their discharge.

**Figure 13 bioengineering-10-00624-f013:**
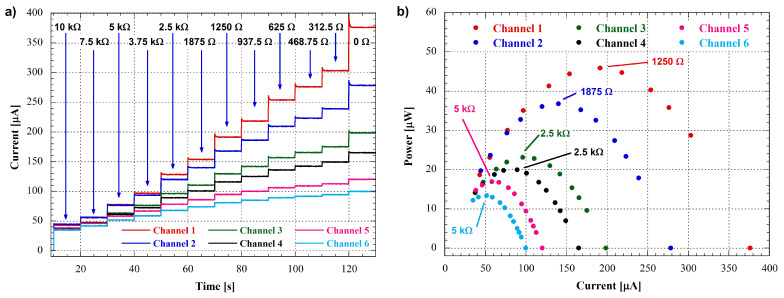
(**a**) Channels’ output current as a function of time; the resistance values indicated in the graph correspond to the selected load resistances of the digital potentiometer. (**b**) Battery packs’ output power as a function of output current; resistances reported in the graph represent the values closest to the peak of the curve among those selected to perform the power measurement.

**Table 1 bioengineering-10-00624-t001:** Analysis of the collected data for the output resistance evaluation.

Channel	$A_{2}$ [V]	Inferred $R_{\mathrm{out}}$ [kΩ]	Nominal $R_{\mathrm{out}}$ [kΩ]	% Error
1	0.48772	1.2961	1.32	−1.8%
2	0.52775	1.8953	1.82	+4.1%
3	0.46577	2.3461	2.52	−6.9%
4	0.48623	2.9460	3.02	−2.5%
5	0.56311	4.5856	4.22	+8.6%
6	0.53626	5.3771	5.02	+7.1%

## Data Availability

Not applicable.

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
