# Peer review of "Customized Multichannel Measurement System for Microbial Fuel Cell Characterization"

_bioengineering, 2023, doi:10.3390/bioengineering10050624_

Round 1

Reviewer 1 Report

This manuscript reports the Customized Multichannel Measurement System for Microbial Fuel Cell Characterization. This work is interesting and well-presented however, following revisions should be made before publication:

1.      There are many typos and grammar errors. The manuscript should be checked and revised carefully.

2. the literature review is insufficient. following articles based on energy applications can be used as reference :doi.org/10.1016/j.est.2023.106713, doi.org/10.1016/j.compositesb.2023.110558

Reviewer 2 Report

In this paper, the development of an automatic and customized measuring system for precise measurements of voltage-current signals generated by MFCs are reported. The system can perform multi-step discharge protocols to accurately measure the power output of MFC. The functionality of the system was tested using a six-channel setup, and the results demonstrated its ability to detect and distinguish current signals. Overall, the developed measuring system is a useful tool for characterizing the performance of MFCs.

The content of the manuscript fits the research scope of the journal, and will also arouse the wide reading interest of the journal readers. However, the manuscript needs some improvements. Therefore, I decided to give a minor revision, hoping that the author can make targeted improvement and promotion according to the suggestions put forward by the reviewer.

1. The quantity of the keywords should be enough to attract readers' interest in reading. Currently, there are few keywords and there is no good summary of the content of the manuscript. Therefore, I suggest adding keywords appropriately, such as 'sigma-delta analogue-to-digital converter', 'trans-impedance amplifier', and 'dual-channel board'.

2. Page 1, 'Microbial Fuel Cells (MFCs) represent a promising technology for sustainable energy generation [1–4].' This should be compared to traditional fossil fuels, MFCs are more sustainable. Compared with hydrogen energy, solar energy, and wind energy, the sustainability of MFCs is not so obvious.

    It can be introduced briefly, for example, the destruction of the natural environment and damage to human health caused by greenhouse gases, toxic smoke, and dust due to the use of fossil energy is increasingly significant. There is an urgent need to replace traditional fossil fuels with renewable and sustainable energy sources [Energies 2023, 16(4), 1653]. 

3. Page 1, 'As a result, MFCs can generate electrical power continuously, without the need for external energy sources, such as sunlight or wind.' When mentioned 'continuously', the drawback of sunlight or wind should be highlighted to make the significance of MFCs more obvious. The unique intermittence and instability of solar/wind energy have brought major challenges to the stable operation of the power system, opening temporal and spatial gaps between the consumption of the energy by end-users and its availability [10.3390/batteries8110202].

4. Page 2, 'Alternatively, MFCs can produce bio-hydrogen instead of electricity, which can be used as a fuel or feedstock for other industrial processes [38–40].' Since PEMFCs based on hydrogen are also very popular today, can the bio-hydrogen be used in PEMFCs directly as the hydrogen source? This may be more interesting to the readers, and some possibilities can be found in 10.1016/j.ijhydene.2013.03.112 

5. Page 6, although many voltages and current calculation equations are introduced here, and many specific figures are given, what are the maximum current density and maximum voltage for MFC single-cell operation? This value has not been given and should be added since the readers are not well aware of the operating parameters of the MFC. Of course, the common power density of MFCs should also be provided if possible.

6. Page 10-14, what kind of material is used in the MFC used for testing? For example, what is the original material for ion exchange membrane and energy generation? These should be clearly described in the text. Otherwise, the measurement significance of this MFC is very limited. Are widely used or commercial membrane materials used? Such as perfluorosulfonic acid membrane-Nafion, which contains hydrophobic PTFE backbone and hydrophilic -SO3H groups  (10.1016/j.ssi.2018.01.038).

7. Page 14, 'As shown in Table 1, the errors that we found are all lower than ±10%. These measurements and elaborations demonstrate the ability of the measuring system to accurately measure power output from MFCs. The consistent results across all six channels further attest to the precision and reliability of the measuring system.'  Although the authors claim that the average error is less than 10%, the authors did not compare this result with the error values of other commercial products. I think this practice detracts from the significance of the methods described in the manuscript, and it is easy for readers to think "the author is talking to himself" in the absence of comparison.

For grammar issues, it is suggested that the author double-check the small grammar errors in the full text, especially the lack of and redundant use of definite articles.
